# Two-Year Follow-Up Shows Gentamicin-Coated Tibial Nails Reduce Infection Rates in Open Tibial Fractures

**DOI:** 10.3390/antibiotics14060532

**Published:** 2025-05-22

**Authors:** Álvaro I. Zamorano, Matías A. Vaccia, Carlos F. Albarrán, Rodrigo I. Parra, Tomás Turner, Ignacio A. Rivera, Tomás Errázuriz, Andrés Oyarzún, Osvaldo A. Garrido, Pablo F. Suárez, Pierluca Zecchetto, Luis A. Bahamonde

**Affiliations:** 1Orthopedics and Traumatology Service, Hospital Clínico Universidad de Chile, Santiago 8380456, Chile; mvaccia@hcuch.cl (M.A.V.); carlos.albarranr@gmail.com (C.F.A.); asaoyarzunm@gmail.com (A.O.); lbahamondemunoz@gmail.com (L.A.B.); 2Lower Extremities Trauma Unit, Hospital Clínico Mutual de Seguridad, Santiago 9190015, Chile; oagarrido@gmail.com (O.A.G.); pfsuarezs@yahoo.com (P.F.S.); p.zecchetto@gmail.com (P.Z.); 3Postgraduate School Orthopedics and Traumatology Department, University of Chile, Santiago 8330111, Chile; rodrigoparram95@gmail.com (R.I.P.); tomasturner@ug.uchile.cl (T.T.); ignacioandresrivera@gmail.com (I.A.R.); tomaserra97@gmail.com (T.E.); 4Department of Orthopedics and Traumatology, McMaster University Medical Center, Hamilton, ON L8N 3Z5, Canada

**Keywords:** tibial fractures, open fractures, fracture-related infections, gentamicin, intramedullary nails, biofilm

## Abstract

**Introduction**: Open tibial fractures carry a high risk of fracture-related infection (FRI), and prevention typically relies on early antibiotics and debridement. However, achieving optimum local antibiotic concentration remains challenging. Gentamicin-coated intramedullary nails (GCN) have been developed to prevent biofilm formation, showing short-term efficacy without interfering with fracture healing. Medium- and long-term data on GCN use are limited. This study aimed to assess the effectiveness and safety of GCN in medium-term follow-up. **Methods**: A prospective cohort study of patients with open tibial fractures was treated with GCN under a standardized protocol, with a minimum follow-up of 24 months. Patients with traumatic amputations, protocol infringement, or loss of follow-up were excluded. The analysis assessed overall FRI incidence by Gustilo–Anderson (GA) classification. **Results**: Of 907 patients, 139 were included, with 2 lost to follow-up. The overall FRI incidence was 8.8%, the average healing time was 34.3 weeks, and the non-union rate was 2.2%. FRI incidence by GA classification was 0% in GA I, 2.9% in GA II, 2.9% in GA IIIA, 44.4% in GA IIIB, and 33.3% in GA IIIC. External fixation (EF) was required in 45.2% of cases, with 16.1% developing FRI (14.3% in GA II, 2.8% in GA IIIA, 50% in GA IIIB, and 33.3% in GA IIIC). In non-EF cases, FRI occurred in 2.7% of patients (2.9% in GA IIIA and 25% in GA IIIB). No adverse effects were reported due to locally administered gentamicin. **Conclusions**: In the medium term, GCN has consistently demonstrated safety and efficacy in preventing FRI in open tibial fractures, particularly in GA IIIA cases, even with the use of temporary EF. These findings highlight its potential as a valuable tool in managing open tibial fractures. However, further studies with long-term outcomes are needed to evaluate its effectiveness in GA IIIB and IIIC fractures.

## 1. Introduction

The tibia is the bone most frequently involved in open fractures of the skeleton, significantly increasing the risk of fracture-related infection (FRI) compared with closed fractures [1]. Several risk factors for infection have been identified, including the extent of bony and soft tissue damage (stratified by the Gustilo–Anderson [GA] classification), high-energy trauma, fractures in polytraumatized patients, temporary use of an external fixator before definitive osteosynthesis, obesity, diabetes mellitus, tobacco use, and other comorbidities [2].

An FRI incidence of 12.2 per 100,000 person-years was reported following open tibia fractures [3]. Currently, FRI is considered a priority pathology in both developed and developing countries, yet there is a lack of data on microbiology, antibiotic resistance, and effective antibiotic coverage [3].

FRI management presents substantial challenges, often necessitating prolonged antibiotic use, extended hospital stays, and multiple surgical interventions [4]. The recently described “triangle of death”, encompassing biofilm formation, antibiotic resistance, and infection, highlights the major factors contributing to both the development of FRI and the difficulty of its treatment [5]. This complication increases lost workdays sixfold and healthcare costs fivefold, underscoring the importance of prevention strategies, which are more cost-effective than treatment [6].

Standard treatment for open tibial fractures typically involves early intravenous antibiotic therapy, surgical irrigation and debridement, and fracture fixation, with intramedullary nailing being the gold standard [1]. However, achieving adequate local antibiotic concentrations—particularly at the bone–implant interface—remains a persistent challenge due to poor local vascularity and the high incidence of antibiotic resistance [5]. A proposed strategy to address this issue is the use of antibiotic-coated intramedullary nails [7], which demonstrated effectiveness and safety in a prospective cohort study with at least 12 months of follow-up. In this study, the use of a gentamicin-coated nail (GCN) was associated with a statistically significant reduction in the incidence of FRI compared to non-gentamicin-coated nails [8].

The Expert Tibial Nail (ETN) PROtect™ (Synthes GmbH, Oberdorf, Switzerland) is a cannulated tibial nail made of a titanium alloy (titanium, 6% aluminum, and 7% niobium) coated with a fully resorbable 10 μm poly(D,L-lactide) (PDLLA) layer containing gentamicin sulfate [9]. In vivo and in vitro studies have demonstrated that the highest gentamicin release occurs within the first hour after implantation, with endosteal levels remaining antimicrobial for the first four hours [10]. Eighty percent of the gentamicin is released within the first 48 h [11], yet it remains detectable at the endosteum for as long as 42 days [10].

Gentamicin, an aminoglycoside, possesses ideal characteristics for local use in the musculoskeletal system, including thermal stability, water solubility, and a broad spectrum of activity, without significantly increasing antibiotic resistance [12]. Its spectrum of activity includes nearly all Gram-negative bacteria, as well as Gram-positive bacteria such as *Staphylococcus aureus* and *Staphylococcus epidermidis*, the most common microorganisms involved in orthopedic infections [13].

The primary adverse effects associated with systemic gentamicin use include acute kidney injury and auditory dysfunction, particularly vestibular damage. Hearing loss most commonly affects higher frequencies. A rare but serious adverse effect is neuromuscular blockade, which is observed primarily in patients with preexisting neuromuscular conditions, such as myasthenia gravis, or in those taking drugs that interfere with neuromuscular transmission [14].

All these adverse effects associated with gentamicin have been reported with systemic use. However, despite the high local concentrations following GCN implantation, systemic gentamicin levels remain below the lowest detectable threshold of 0.2 mg/dL [15]. Furthermore, studies have shown that gentamicin does not interfere with bone healing or cause kidney injury [16].

FRI is a devastating complication following a fracture, and its management continues to pose significant challenges for both patients and healthcare providers. Various strategies for prevention and treatment have been developed, including the use of antibiotic-coated intramedullary nails. Specifically, GCNs have demonstrated early benefits and safety in the management of open tibial fractures. However, to date, no studies have evaluated whether their efficacy and safety are maintained in the medium term.

The main objective of this study is to extend the follow-up period to at least two years for patients with open tibial fractures treated with GCNs, as previously reported, and to evaluate the maintenance of its effectiveness over the medium term. The study also aims to assess these outcomes independent of GA classification, ensuring that fracture consolidation rates are not adversely affected by high doses of local antibiotics.

## 2. Results

Out of 907 tibial fractures (both closed and open), 139 patients with open tibial fractures treated with GCN during the defined period met the inclusion and exclusion criteria. Two additional patients were excluded due to loss to follow-up before fracture healing, resulting in a final cohort of 137 patients.

The statistical analysis of demographic variables (Table 1) showed an average age of 31.7 years with a standard deviation (SD) of 14.4 years. By age group, the most commonly affected were patients aged 18 to 30 years (45 patients) and 31 to 40 years (36 patients). A clear male predominance was observed, accounting for 92.7% of cases. Regarding comorbidities, the cohort had a smoking prevalence of 13.3% and a diabetes mellitus prevalence of 10.2%.

The most frequent mechanism of injury was motor vehicle collision (MVC) in 36 patients (26.3%), followed by pedestrians struck by vehicle (11 patients, 8.02%), and fall from height (4 patients, 2.9%). Other mechanisms, such as ground-level falls, direct blows, and crush injuries, had even lower incidences (Figure 1).

Fracture morphology, classified using the AO/OTA system, demonstrated a high incidence of high-energy fractures (Table 2), particularly AO 42B2 (21.3%), followed by 42B3 (10.3%), 42C2 (8.8%), and 42C3 (7.4%). The middle third of the tibia (AO 42) was the most frequently affected segment, with significantly lower incidences in the proximal and distal thirds.

The average time to fracture healing was 34.3 weeks (SD 23.8). Stratified by GA classification, type I fractures healed in an average of 21.7 weeks (SD 5.6), type II fractures in 24.6 weeks (SD 24.6), type IIIA fractures in 32 weeks (SD 15.7), type IIIB fractures in 63.5 weeks (SD 37.7), and type IIIC fractures in 91.7 weeks (SD 45).

The overall non-union rate was 2.2%. By GA classification, there was one non-union (1.4%) in type IIIA fractures and two non-unions (11.1%) in type IIIB fractures. No non-unions were observed in other GA subgroups.

The overall incidence of FRI in the cohort was 8.8%. By GA classification, FRI incidence was 0% in GA I fractures, 2.9% (1/34) in GA II fractures, 2.9% (2/70) in GA IIIA fractures, 44.4% (8/18) in GA IIIB fractures, and 33% (1/3) in GA IIIC fractures (Table 3).

Regarding external fixation (EF) use, 62 patients (45.3%) underwent temporary EF stabilization, while 75 patients (54.7%) had definitive fixation during the initial surgical procedure (Table 4). The main indications for EF were polytrauma (19 cases, 30.6%), initial management at another hospital with subsequent transfer for definitive care (38 cases, 61.3%), and severe soft tissue damage contraindicating immediate definitive fixation (5 cases, 8%).

The incidence of FRI in the EF group was 16.1% compared to 2.7% in the non-EF group. Stratified by GA classification, the EF group had one FRI case (14.3%) in type II fractures, one case (2.8%) in type IIIA fractures, seven cases (50%) in type IIIB fractures, and one case (33.3%) in type IIIC fractures. In the non-EF group, there was one FRI case (2.9%) in type IIIA fractures and one case (25%) in type IIIB fractures, with no infections in other subgroups.

During follow up, there were no cases of kidney injury or ototoxicity suspicious of being secondary to the local use of gentamicin in the implants.

## 3. Discussion

The initial management of open tibia fractures focuses on mitigating the risk of FRI through timely intravenous antibiotics, surgical irrigation and debridement, fracture stabilization, and prompt soft tissue coverage [17]. Even with all these measures, open tibia fractures carry a substantial FRI risk, with rates reported as high as 52% [18]. Once FRI develops, it leads to significant healthcare costs, reduced patient quality of life, and persistently high amputation and recurrence rates [5,6].

Giannoudis and Giordano recently described the “triangle of death” in FRI management, emphasizing the importance of achieving adequate local antibiotic concentrations. The tibia’s poor vascularization contributes to the difficulty in preventing biofilm formation, which is a major factor in infection persistence and recurrence. Biofilm acts as both a protective barrier and a growth medium for microbes, rendering systemic antibiotic treatments less effective [19,20].

Preventive strategies have been explored across preoperative, intraoperative, and postoperative phases. Among intraoperative measures, the combined use of systemic and local antibiotics effectively reduces early biofilm formation [21]. Techniques such as vancomycin powder application, antibiotic hydrogels, and antibiotic-coated implants—such as gentamicin-coated nails (GCNs)—have gained significant attention for their potential to lower infection rates [19].

The present study evaluated a predominantly working-age population with a clear male predominance (92.7%), consistent with demographics reported in similar studies [22]. Tobacco use (13.3%) and diabetes mellitus (10.2%) were common comorbidities, both of which are known risk factors for FRI. The high-energy nature of the injuries in this cohort is evidenced by the large proportion (66.4%) of GA III fractures, highlighting the complexity of these cases.

Khoury et al., in a retrospective study of 912 open tibial fractures, reported an overall FRI incidence of 16%, with rates stratified by GA classification as follows: 8% for type I–II fractures, 15% for type IIIA fractures, 35% for type IIIB fractures, and 36% for type IIIC fractures [23]. Polymicrobial infections accounted for 20–35% of cases, especially in open fractures [24,25].

In this prospective cohort of open fractures treated exclusively with GCNs, the overall FRI rate was 8.8%, lower than previously reported rates. Subgroup analysis by GA classification revealed further reductions compared to the Khoury et al. series, particularly in GA I–II fractures (0.7%) and GA IIIA fractures (2.9%). However, FRI rates remained high in GA IIIB (44.4%) and IIIC (33%) fractures, likely due to delays in definitive soft tissue coverage by a specialized unit.

A known risk factor for FRI is the use of external fixators (EF). Bunzel et al. reported that EF use in low-grade open and closed fractures significantly increases FRI risk, particularly with prolonged use (>15 days) [26]. Consistent with their findings, this study showed a higher FRI incidence (14.3%) in GA II fractures treated with EF compared to 0% in those managed with definitive fixation during the initial surgery. In other GA groups, no significant differences were observed, possibly due to the short duration of EF use before definitive fixation. Another explanation could be that the high antibiotic concentrations achieved after GCN implantation effectively treat the wound contamination associated with EF pins. Further research is warranted to explore whether GCNs reduce FRI incidence in patients stabilized with EF, both in open and closed tibial fractures.

Zamorano et al. demonstrated the safety and efficacy of GCNs in preventing FRI up to one year postoperatively [8], while Leal et al., in a systematic review and meta-analysis, observed a trend favoring GCNs in reducing FRI rates in open tibial fractures, although statistical significance was not achieved [22]. Notably, Zamorano et al.’s study—the largest series to date—was not included in Leal et al.’s analysis. The findings of these studies as well as those reported by the cohort presented in this paper suggest that GCNs are particularly effective in lower-energy injuries (GA I–IIIA), whereas outcomes in severe injuries (GA IIIB–IIIC) may rely more heavily on other factors, such as timely soft tissue coverage as part of an orthoplastics approach.

This study, which includes a large cohort with at least two years of follow-up, confirms that GCNs effectively prevent biofilm formation and maintain their safety profile in the medium term. There were no clinical or laboratory signs of gentamicin-related adverse effects, such as ototoxicity or nephrotoxicity.

Considering the results of this study and the literature published to date, the use of GCNs in open tibial fractures has been demonstrated to be both safe and effective in the short and medium term. This group believes that the significant reduction in FRI incidence (particularly in GA I to IIIA fractures), coupled with the absence of adverse effects associated with this implant, supports its adoption as the implant of choice for open tibial fractures.

It is important to consider the cost of this implant. As of 2020, the cost of a GCN in Chile was USD 1767, while a non-GCN was USD 1514, making GCNs 16% more expensive than conventional nails. This cost difference remains similar in 2025, with a difference of 22%. A future cost-effectiveness study is needed to determine whether the reduction in FRI incidence justifies the widespread use of GCNs from an economic perspective despite their slightly higher cost.

The study’s strengths include its prospective design, long follow-up period, and the comprehensive care afforded by a workers’ compensation system. However, the study is limited by its descriptive nature, lack of a control group, and the challenges of managing severe injuries (GA IIIB–IIIC) due to limited access to microsurgery-trained units for early soft tissue coverage.

There are few plastic surgeons trained in reconstruction in Chile and even fewer trained in microsurgery, which inevitably leads to delays in complex soft tissue coverage, particularly in high-energy fractures. These factors contributed to a delay of more than seven days in definitive coverage for this subset of patients, a known risk factor for FRI development.

## 4. Methods

This prospective descriptive study included 907 patients with tibial fractures managed by the Lower Extremity Trauma Unit at the Hospital Clínico Mutual de Seguridad (HCMS) between January 2018 and September 2022. Patients were included if they underwent definitive fixation with a gentamicin-coated intramedullary nail (GCN), with at least the definitive fracture fixation performed by the Lower Extremity Trauma Unit, and had a minimum follow-up of 24 months.

Exclusion criteria included the following:-Closed tibial fractures;-Definitive fracture fixation performed at other facilities, with subsequent referral to HCMS for follow-up or treatment of complications;-Definitive osteosynthesis without an intramedullary nail;-Use of non-GCN implants in patients with open tibial fractures;-Follow-up period of less than 24 months;-Loss to follow-up for any cause;-External fixation (EF) maintained for more than three weeks due to the increased risk of FRI;-Soft tissue coverage performed more than 48 h after fracture fixation due to increased FRI risk;-Flap necrosis;-Amputation;-Segmental bone stock loss requiring multiple surgeries and reconstruction techniques.

The open fracture management protocol involved administering intravenous first-generation cephalosporins for GA type I fractures, with gentamicin added for GA types II and III. Antibiotics were administered within three hours of the injury. For GA type I fractures, a single dose of intravenous antibiotics was administered, while three doses were given for GA type II and IIIA fractures. For GA types IIIB and IIIC fractures, antibiotics were continued until adequate soft tissue coverage was achieved. Intravenous penicillin was added for cases involving gross contamination, stagnant water, or agriculture-related accidents.

Surgical irrigation and debridement were performed within 24 h. Fracture fixation with a temporary EF or definitive intramedullary nailing (with a GCN) was based on injury severity, including associated injuries, soft tissue damage, and gross contamination. Polytraumatized patients were assessed in collaboration with intensive care unit (ICU) specialists and anesthesiologists to determine whether they were candidates for damage control surgery or early definitive fixation. When necessary, temporary EF was maintained until patients were cleared for definitive surgery.

For cases involving local soft tissue or vascular damage, consultations with plastic and vascular surgeons were conducted to perform early soft tissue coverage or vascular repair by trained specialists. Definitive surgery with GCN was the standard protocol unless prolonged EF use was required.

Data were collected prospectively during the follow-up period for all patients meeting the inclusion and exclusion criteria. The implant used in all cases was the GCN (Expert Tibia Nail PROtect™, Synthes GmbH, Oberdorf, Switzerland).

FRI was defined per the 2018 FRI consensus [27]. Infection was confirmed in the following scenarios:-Open tibial fracture requiring surgical debridement due to persistent or purulent drainage, exposed hardware, or exposed bone;-At least two positive intraoperative cultures of bone or deep tissue samples isolating the same infectious microorganism.

Fractures that did not meet the FRI criteria were considered not infected and regarded as successful treatments. Conversely, any fracture meeting the FRI criteria was classified as a treatment failure.

Complete bone healing was defined as a RUST scale score [28,29] of at least 9 points [30]. Healing was assessed at the time of medical discharge. Comprehensive follow-up was ensured for the study population, with evaluations by internal medicine and otorhinolaryngology consultants conducted as needed to address complications. Preoperative studies, including plasma creatinine levels, were reviewed in cases requiring secondary interventions for FRI or non-union.

The analysis included demographic characteristics of the sample, expressed as percentages of the total population and as averages for age. For the primary study objective, the incidence of FRI was calculated for the entire cohort and stratified by GA classification as well as by the use of EF. The incidence for each subgroup was calculated with a 95% confidence interval.

This statistical approach was chosen because the study is observational and lacks a control group for comparison. The method allowed for characterization of the sample within the follow-up time frame included in the study.

## 5. Conclusions

The use of gentamicin-coated intramedullary nails (GCNs) in open tibial fractures demonstrates consistent safety and efficacy in the medium term, particularly in GA IIIA fractures, even in cases requiring temporary external fixation. However, their effectiveness in higher-grade fractures (GA IIIB and IIIC) warrants further investigation, particularly when evaluated in centers with prompt access to an orthoplastic approach.

Randomized controlled trials are needed to confirm the findings of this study and the existing literature, as high-quality evidence on the effectiveness of this implant remains limited. Future research should prioritize the development of comprehensive management protocols, including adherence to international guidelines on timing for soft tissue coverage, and explore adjunctive strategies to improve outcomes in these complex and challenging cases.

## Figures and Tables

**Figure 1 antibiotics-14-00532-f001:**
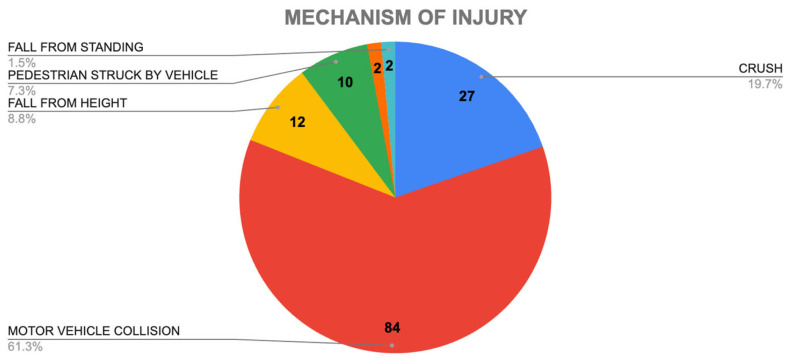
Cohort mechanism of injury demographics, highlighting that 61.3% of injuries corresponded to motor vehicle collision.

**Table 1 antibiotics-14-00532-t001:** Demographic and risk factors variables. Distribution of participants sex, mean age per group, proportion of participants in each age, the use of tobacco, and the presence of diabetes per sex.

	Male	Female
Sex	92.7%	7.3%
Mean Age	31.1	40.3
Age group		
18–30	41	4
30–40	34	2
40–50	16	0
50–60	20	3
60–70	14	1
70–80	1	0
80–90	1	0
	Yes	No
Tobacco use	13.3%	86.7%
Diabetes	10.2%	89.8%

**Table 2 antibiotics-14-00532-t002:** This table summarizes the fracture demographics based on the AO/OTA Classification System, showing the number and percentage of fractures within each classification, most of them being 42.

Classification	Number	Percentage
41A1	0	0%
41A2	1	0.7%
41A3	5	3.7%
42A1	10	7.4%
42A2	13	9.6%
42A3	29	21.3%
42B2	29	21.3%
42B3	14	10.3%
42C2	12	8.8%
42C3	10	7.4%
43A1	2	1.5%
43A2	4	2.9%
43A3	5	3.7%
43C1	1	0.7%
43C2	1	0.7%
43C3	1	0.7%

**Table 3 antibiotics-14-00532-t003:** Infection rate, non-union, and healing time by Gustilo–Anderson classification, with GA IIIA representing 51% of patients. Overall infection rate for open fractures was 8.8%, with an overall non-union rate of 2.2%.

2 Years	Number	Percentage	Infection	Infection Rate	Healing Time (Weeks)	Non-Union	Non-Union Rate
Total	137						
Open Fracture	137		12	8.8%	34.3	3	2.2%
GA I	12	8.76%	0	0.0%	21.7	0	0.0%
GA II	34	24.82%	1	2.9%	24.6	0	0.0%
GA IIIA	70	51.09%	2	2.9%	32	1	1.4%
GA IIIB	18	13.14%	8	44.4%	63.5	2	11.1%
GA IIIC	3	2.19%	1	33.3%	91.7	0	0.0%

**Table 4 antibiotics-14-00532-t004:** This table shows infection rates for open fractures based on whether external fixation (Ex Fix) was used or not. The infection rate was 16.1% for patients who underwent external fixation and 2.7% for those who did not undergo external fixation.

2 Years	Ex Fix	Infection	Infection Rate	No Ex Fix	Infection	Infection Rate
Total						
Open Fracture	62	10	16.1%	75	2	2.7%
GA I	2	0	0.0%	10	0	0.0%
GA II	7	1	14.3%	27	0	0.0%
GA IIIA	36	1	2.8%	34	1	2.9%
GA IIIB	14	7	50.0%	4	1	25.0%
GA IIIC	3	1	33.3%	0	0	0.0%

## Data Availability

Data are contained within the article.

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
