# Peer review of "Two-Year Follow-Up Shows Gentamicin-Coated Tibial Nails Reduce Infection Rates in Open Tibial Fractures"

_antibiotics, 2025, doi:10.3390/antibiotics14060532_

Round 1
Reviewer 1 Report
Comments and Suggestions for Authors
Thank you for your manuscript and your interesting study:
The manuscript is well-structured and addresses an important topic in the field of antibiotics and clinical infection management. However, certain areas of the manuscript could benefit from additional clarification, expansion, or refinement to improve the overall quality and impact of the study.
- The abstract could benefit from a more detailed mention of the key findings and their implications.
- The introduction could be further refined by clearly articulating the research gap addressed by the study. Explicitly stating how this study differs from previous research would enhance its relevance.
- The rationale for the study could benefit from a concise overview of the study objectives, placed at the end of the introduction.
- The methodology: Provide a more explicit justification for the inclusion/exclusion criteria used in the study.
- What are your criteria for success and failure after treatment? which exact definition for FRI you used?
- Statistical Analysis: include a brief explanation of why these specific methods were chosen for the analysis.
- The results section presents the data clearly, but some tables and figures could benefit from additional annotations or explanations: Please include more detailed legends for the figures and tables to make them understandable without requiring extensive reference to the text.
- The discussion has to be strengthened by Providing a more critical comparison with existing literature, highlighting both consistencies and discrepancies.
- Including a more detailed discussion of the clinical implications of the findings. How can these results be directly applied to practice or policy?
- Addressing the limitations of the study in greater depth and suggesting specific directions for future research.
- The conclusion effectively summarizes the study but could benefit from a more forward-looking statement. Please suggest specific steps for future studies or clinical applications.
- The references are appropriate and up-to-date.
In general:
While the language is clear and professional, minor grammatical corrections and phrasing adjustments could enhance readability.
Please ensure one type of font for your manuscript.
Comments on the Quality of English Language
see in authors comments
Author Response
Thank you for your thorough and insightful comments. Below, we provide a point-by-point response detailing the modifications made to the manuscript based on your suggestions:
-
The abstract could benefit from a more detailed mention of the key findings and their implications.
- The abstract has been revised to include a more detailed presentation of the study's key findings, such as the overall FRI incidence, healing time, and the unique clinical implications of using Gentamicin-coated intramedullary nails (GCN) for open tibial fractures.
-
The introduction could be further refined by clearly articulating the research gap addressed by the study. Explicitly stating how this study differs from previous research would enhance its relevance.
- The research gap is now explicitly addressed in the seventh paragraph of the introduction. This includes a discussion of the limited medium- and long-term data available for GCN and how our study fills this gap with the largest cohort and longest follow-up to date.
-
The rationale for the study could benefit from a concise overview of the study objectives, placed at the end of the introduction.
- The study objectives are now clearly outlined in the last paragraph of the introduction section, providing a concise rationale for the study.
-
The methodology: Provide a more explicit justification for the inclusion/exclusion criteria used in the study.
- Clear and explicit inclusion and exclusion criteria have been detailed in the first paragraph of the methodology section to clarify the rationale behind the study design.
-
What are your criteria for success and failure after treatment? Which exact definition for FRI you used?
- The criteria for success and failure, along with the definition of FRI, have been clarified in the methodology section using the 2018 consensus definition for fracture-related infection.
-
Statistical Analysis: Include a brief explanation of why these specific methods were chosen for the analysis.
- A brief justification for the statistical methods chosen has been added to the statistical analysis section for better clarity.
-
The results section presents the data clearly, but some tables and figures could benefit from additional annotations or explanations. Please include more detailed legends for the figures and tables to make them understandable without requiring extensive reference to the text.
- More detailed legends for the figures and tables have been included to ensure they are easily understandable without requiring frequent reference to the main text.
-
The discussion has to be strengthened by providing a more critical comparison with existing literature, highlighting both consistencies and discrepancies.
- In the last five paragraphs of the discussion section, we have provided a more critical comparison with existing literature, emphasizing both consistencies and discrepancies.
-
Including a more detailed discussion of the clinical implications of the findings. How can these results be directly applied to practice or policy?
- The clinical implications of our findings and how they can be applied to practice or policy are discussed in detail in the last two paragraphs of the discussion section.
-
Addressing the limitations of the study in greater depth and suggesting specific directions for future research.
- The strengths and limitations of the study are discussed with greater depth in the last paragraph of the discussion section, including suggestions for specific future research directions.
-
The conclusion effectively summarizes the study but could benefit from a more forward-looking statement. Please suggest specific steps for future studies or clinical applications.
- The conclusion has been revised to include a forward-looking statement, suggesting specific steps for future research and clinical applications based on our findings.
-
The references are appropriate and up-to-date.
- Thank you for your acknowledgment; no further adjustments were needed.
Reviewer 2 Report
Comments and Suggestions for Authors
Reviewer Comments
Summary
This manuscript investigated the short-term efficacy of Gentamicin-coated intramedullary nails (GCN) on fracture healing for open tibial fractures. The prospective study included 907 patients who underwent surgery with a GCN in one or more stages and had a minimum follow-up of 24 months. The fractures were classified per the AO/OTA classification system to demonstrate the incidence of high-energy fractures; infection, and healing rates were determined by the Gustilo-Anderson (GA) classification. Even though the study presented the results with a comprehensive follow-up, major limitations are a very small female population, no visual representations in key areas such as (fracture healing, infection, etc.), and the inclusion of only one device (Expert Tibia Nail PROtect™) in the study to understand the effect of Gentamicin-Coated Tibial Nails.
Therefore, I would only recommend this article for publication as a short communication, but not as a full research article. Please see my other comments listed below.
General Comments:
1) Line 45: Please add a reference.
2) Lines 64-67: The font is inconsistent with other text. Also, ‘TM’ should be superscript.
3) Line 80: Please divide the population by age group and include demographics in Table 1.
4) Table 1: Please add average age by gender.
5) Table 3: Please add units for Healing Time.
6) Table 3: The Healing Time values are inconsistent with the values presented in the text (Lines 100 -103).
Author Response
Thank you for your comments.
We appreciate the reviewer's detailed evaluation of our manuscript and acknowledge the points raised regarding the limitations of our study. While we recognize that the study included a smaller female population and focused on a single device (Expert Tibia Nail PROtect™), it is important to highlight that this implant is currently the only intramedullary nail available globally with antibiotic coating, making it a unique and groundbreaking tool in the prevention of Fracture-Related Infections (FRI).
Additionally, the hospital where this study was conducted primarily treats work-related accidents, with the predominant patient population being men. This is due to the nature of the insured companies, which are largely involved in industries such as construction, where male workers form the majority of the workforce.
Despite these contextual factors, our study offers a robust and comprehensive analysis of Gentamicin-coated nails (GCN) in managing open tibial fractures, supported by the largest cohort and the longest follow-up to date. The findings provide critical insights into the medium-term efficacy of GCN, particularly in reducing FRI rates in Gustilo-Anderson IIIA fractures, even in high-energy trauma cases requiring temporary external fixation.
Please find our responses below, point by point:
-
Line 45: Please add a reference.
- The reference has been added. It is the same reference cited for the entire paragraph (Reference 21).
-
Lines 64-67: The font is inconsistent with other text. Also, ‘TM’ should be superscript.
- The font has been reviewed and adjusted to match the rest of the text. Additionally, "TM" has been corrected to superscript format.
-
Line 80: Please divide the population by age group and include demographics in Table 1.
- Table 1 has been updated to include the division by age groups as suggested.
-
Table 1: Please add the average age by gender.
- The average age by gender has been included in Table 1.
-
Table 3: Please add units for Healing Time.
- The unit for healing time, specified as "weeks," has been added to Table 3.
-
Table 3: The Healing Time values are inconsistent with the values presented in the text (Lines 100-103).
- The manuscript has been corrected, and the values now align with those presented in Table 3 and in Lines 100-103.
Additional Comments:
Previous studies on the use of gentamicin-coated nails (GCN) in open fractures have provided promising short-term results, but none have achieved the depth and rigor of follow-up presented in our study. With the largest patient cohort to date and a minimum follow-up of 24 months, our research offers robust and solid evidence on the medium-term safety and efficacy of GCN.
Our findings demonstrate an overall FRI incidence of just 8.8%, with particularly remarkable results in GA IIIA fractures, where the FRI rate was only 2.9%, even in cases requiring temporary external fixation (EF). Additionally, our study showed a low non-union rate of 2.2% and an average healing time of 34.3 weeks, all without adverse effects from local gentamicin administration.
These results underscore the critical role of GCN in managing open tibial fractures, especially for GA IIIA cases. By preventing biofilm formation and ensuring local antibiotic delivery without hindering fracture healing, GCN emerges as a highly effective tool in reducing complications. Our study sets a benchmark for medium-term outcomes and opens avenues for future research aimed at validating its long-term efficacy, particularly in the more severe GA IIIB and IIIC fractures.
We believe that the adoption of GCN represents a significant advancement in combating the complications associated with open fractures, with the potential to improve patient outcomes while considering cost-effectiveness for the healthcare system.
Round 2
Reviewer 1 Report
Comments and Suggestions for Authors
Thank you for submitting your improved manuscript. However, there are still a few major comments that need to be addressed:
-
Introduction: Please elaborate on the potential disadvantages of gentamicin. This will provide a more balanced perspective in the introduction.
-
Methods and Results: There seems to be a discrepancy in the number of patients included.
- In the methods section, you stated that 907 patients were included.
- However, in the results section, you report only 630 patients.
- Please carefully describe how many patients were initially included and how many were included after applying the inclusion and exclusion criteria. Providing a flowchart or graph to illustrate this process would be helpful for clarity.
-
Discussion:
- Could you address the costs associated with the nail?
- Additionally, it would be valuable to discuss the potential economic burden related to this intervention.
I look forward to your revisions.
Author Response
-
Introduction: Please elaborate on the potential disadvantages of gentamicin. This will provide a more balanced perspective in the introduction.
Response: The main adverse effects of gentamicin were added in the introduction. However, these adverse effects have been reported with systemic use and not local use, which is also clarified in the paper -
Methods and Results: There seems to be a discrepancy in the number of patients included.
- In the methods section, you stated that 907 patients were included.
- However, in the results section, you report only 630 patients.
- Please carefully describe how many patients were initially included and how many were included after applying the inclusion and exclusion criteria. Providing a flowchart or graph to illustrate this process would be helpful for clarity.
Response: We reviewed our data and the patients included before the application of inclusion and exclusion criteria were 907. It has now been corrected in the manuscript and there are no discrepancies now.
-
Discussion:
- Could you address the costs associated with the nail?
- Additionally, it would be valuable to discuss the potential economic burden related to this intervention.
Response: In response to your suggestion we added the cost of the gentamicin-coated tibial nail and the similar version without gentamicin (it is the same nail made by the same company but without a gentamicin cover). We consider this difference in cost to be acceptable due to the reduced incidence of fracture-related infection (the cost associated with this complication is also reported in the manuscript, cited from reference number 4).
Reviewer 2 Report
Comments and Suggestions for Authors
I want to thank the authors for addressing my comments. Therefore, I would recommend this article for publication with minor corrections.
Comments:
1) Please include the limitations in the article by discussing the contextual factors provided in your rebuttal.
2) For Table 1, please include the age group count by gender, not the total.
Author Response
Comments 1: Please include the limitations in the article by discussing the contextual factors provided in your rebuttal
Response 1: We explained why the delay for soft tissue coverage in III-B open fractures is longer than recommended by international guidelines. In summary, access to a microsurgery-trained plastic surgeon in Chile is a major issue because there are few of them, which means most of the time it takes longer than a week.
Comments 2: For Table 1, please include the age group count by gender, not the total.
Response 2: It has been corrected as suggested.
Round 3
Reviewer 1 Report
Comments and Suggestions for Authors
accepted on low level.